# LayeredGS: Efficient Dynamic Scene Rendering and Point Tracking with Multi-Layer Deformable Gaussian Splatting

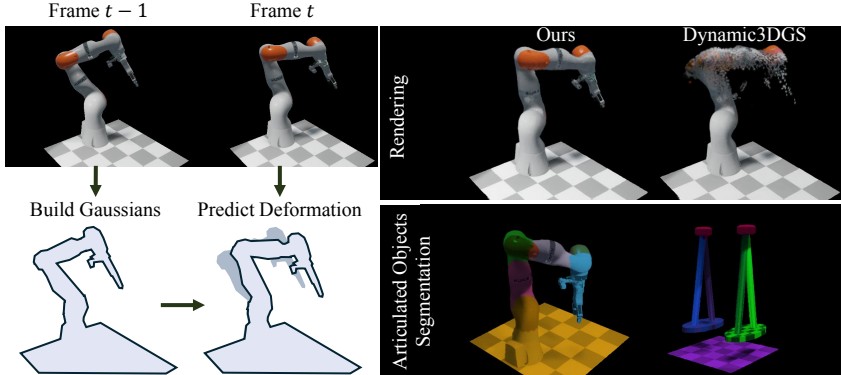

Figure 1: Our method achieves satisfying rendering results being only trained 100 iterations per frame. Leveraging learned deformation information, we also demonstrate successful articulated object segmentation.

## Abstract

Dynamic novel-view synthesis and point tracking have emerged as promising tasks. However, existing methods often struggle with efficiency and accurately capturing deformations. In this paper, we propose LayeredGS, a novel Deformation-based Dynamic Gaussian Splatting method that excels in both 3D tracking of dense scene elements and real-time dynamic scene rendering. By learning Gaussian deformations between frames, LayeredGS preserves their point-like characteristics while capturing motion. Unlike previous methods, our approach optimizes efficiency by grouping Gaussians with similar deformations using a coarse-to-fine clustering structure. Experimental results show the rapid convergence within 100 iterations per time frame on fast-moving dynamic datasets, maintaining rendering quality and tracking accuracy comparable to state-of-the-art methods using only 1/20 training iterations. Additionally, we introduce the application of self-supervised articulated object segmentation, highlighting the utility of deformation information for the first time.

## 1 Introduction

Dynamic novel-view synthesis offers a groundbreaking approach to modeling the 3D world, with promising applications in fields such as AR/VR, robotics, and self-driving cars. The learned dynamic rendering can enable immersive and interactive experiences. In recent years, various attempts (Chen et al., 2022; 2023a; Fridovich-Keil et al., 2022; Hu et al., 2023; Müller et al., 2022; Chen et al., 2023b; Garbin et al., 2021; Hedman et al., 2021; Reiser et al., 2023; Wizadwongsa et al., 2021; Chen et al., 2021; Niemeyer et al., 2022; Wynn & Turmukhambetov, 2023; Yu et al., 2021) have been made to solve this task. These works are inspired by the success of Neural Radiance Field (NeRF) (Mildenhall et al., 2020) and use radiance field to model the 3D scenes. However, the time-consuming network querying and volume rendering procedure make real-time rendering difficult. Moreover, the implicit representation of NeRF limits the possibility of some downstream applications like tracking.

Recently, the emergence of 3D Gaussian Spatting (3DGS) (Kerbl et al., 2023) significantly improves the efficiency of static scene rendering. By modeling a 3D scene as a set of 3D Gaussians and

utilizing efficient rasterization, 3DGS can achieve real-time rendering speed and fast training speed. Such insight also inspires research in dynamic scenes (Wu et al., 2023; Duan et al., 2024; Sun et al., 2024; Luiten et al., 2023). In contrast to implicit approaches that utilize a single module to encode entire dynamic scenes, some explicit online methods (Luiten et al., 2023; Abou-Chakra et al., 2023; Zhang et al., 2023) that try to directly learn the change of position, rotation, and other parameters of each Gaussian between adjacent frames have shown to be more effective in capturing the deformation across time.

These methods (Luiten et al., 2023) can realistically capture the deformation of the scene over time rather than merely accomplishing the rendering task. They preserve the point-like structural characteristics of "3D" Gaussian Splatting, enabling effortless tracking of the evolution of any Gaussian within the dynamic process without the need for any further processing. In particular, Dynamic3DGS (Luiten et al., 2023) uniquely enables 3D tracking across all frames for every scene element, offering the capability to track all Gaussians throughout the entire sequence. This makes it an effective method for tasks requiring fine-grained tracking, but its optimization process is slow. Per-frame optimization often requires a large number of iterations (e.g., 2000 per frame), significantly hindering its use in real-time applications. In addition, simply modeling the translation and rotation of each Gaussian cannot yield good enough results for longer videos, making it difficult to fully utilize the motion information.

These observations motivate us to study for a better deformation strategy. In real-world dynamic scenes, objects are often composed of multiple parts, with Gaussians within each part exhibiting similar deformations. It is thus natural to leverage this *structural* information and reduce the need for training each 3D Gaussian individually. To achieve this, we organize the Gaussians into clusters and optimize the deformation of the entire cluster rather than individual Gaussians. Moreover, the covariance matrix and centroid position of each Gaussian are coupled during deformation. This means we do not need to learn the changes in the covariance matrix and centroid positions separately. By making these changes, we can significantly reduce the number of parameters that need to be optimized, thereby improving the efficiency of the optimization process.

However, the size of the clusters is a trade-off. Training efficiency cannot significantly improve if a cluster contains too few Gaussians. On the other hand, if a cluster contains too many Gaussians, the large cluster tends to move as a whole, making it difficult to model the deformation among the Gaussians within the cluster. This can reduce the modeling capability for objects with detailed movements. Therefore, we adopt a *coarse-to-fine multi-layer cluster structure*. In our experiments, we demonstrate that this approach greatly enhances optimization efficiency, ensuring high rendering quality and 3D tracking performance in complex dynamic scenes. Notably, our method achieves *20× speed up compared with Dynamic3DGS* (Luiten et al., 2023).

A remaining challenge lies in learning the deformation information for each cluster. We address this issue by explicitly constructing a trainable deformation function with parameters that represent a cluster's rotation, translation, and fine-grained scaling. This enables us to learn the deformation information efficiently via backpropagation of a 2D image loss.

Following the acquisition of deformation information, we introduce a straightforward yet important application that is enabled by our method: *Self-supervised* Articulated Object Segmentation. Specifically, we propose clustering the object's parts by our deformation, achieving superior segmentation to real scenes and objects, as shown in Figure 1.

In summary, the contributions of our paper are four-fold:

- We present LayeredGS, a method for online dynamic rendering that achieves both real-time rendering and rapid convergence. LayeredGS delivers rendering quality comparable to state-of-the-art methods, requiring only 1/20 of the iterations and consuming just 1 to 3 seconds per frame for training. Meanwhile, LayeredGS enables 3D point tracking across all frames, providing accurate motion tracking for dynamic scenes.
- We introduce a multi-layer, coarse-to-fine, cluster-based optimization strategy that significantly improves the efficiency of the optimization process.
- We propose a trainable deformation function for clusters, enabling efficient learning of deformation information via backpropagation of a 2D image loss.
- We demonstrate one novel application, Self-supervised Articulated Object Segmentation, showcasing the utility of deformation information *for the first time*.

## 2 RELATED WORKS

**Static Novel-View Synthesis** has become popular in 3D vision in recent years. Specifically, given a set of images from different camera poses, high-fidelity rendered images on novel views are expected. The potential of achieving photorealistic results on this task is revealed by Neural Radiance Field (NeRF) (Mildenhall et al., 2020), which encodes the scene as a fully connected deep network. Following this, a series of works are proposed to improve the efficiency, rendering quality, storage consumption, and other aspects of NeRF (Chen et al., 2022; 2023a; Fridovich-Keil et al., 2022; Hu et al., 2023; Müller et al., 2022; Chen et al., 2023b; Garbin et al., 2021; Hedman et al., 2021; Reiser et al., 2023; Wizadwongsa et al., 2021; Chen et al., 2021; Niemeyer et al., 2022; Wynn & Turmukhambetov, 2023; Yu et al., 2021). However, the design of costly volume rendering and neural networks makes the improvements very challenging, especially in balancing the time efficiency and rendering quality. Recently, 3D Gaussian Splatting (Kerbl et al., 2023) is proposed to elegantly solve this problem by explicit 3D Gaussian representation and differentiable rasterization.

Our work is highly inspired by this but extends from static scenes to dynamic scenes. In particular, we start from static 3D Gaussians and optimize towards the dynamic scene. The natural representation of 3D Gaussians allows for explicit modeling of deformation and high efficiency for both training and inference.

**Dynamic Novel-View Synthesis** is a more challenging task in dynamic scenes. Inspired by the success of NeRF (Mildenhall et al., 2020), various attempts have been made to model the dynamics (Attal et al., 2023; Cao & Johnson, 2023; Fang et al., 2022; Li et al., 2022b;d; 2021; 2023; Park et al., 2021a;b; Pumarola et al., 2021; Fridovich-Keil et al., 2023; Yang et al., 2022; Weng et al., 2022). These works solve the dynamic problem by different routes. Specifically, some works (Li et al., 2022b; Weng et al., 2022; Yang et al., 2022; Zhao et al., 2022) focus on certain scenarios like human motion and leverage prior knowledge, such as human skeletons, to facilitate the synthesis. While achieving impressive results, the modeling strategy cannot be applied to general cases. Deformation-based methods (Attal et al., 2023; Park et al., 2021a;b; Pumarola et al., 2021) build a canonical stage and warp the other frames to this stage. This approach can be applied to more general scenes but can't work well on complex scenes with high variations. Impressed by the high rendering speed of 3DGS (Kerbl et al., 2023), many recent works focus on dynamic scenes with the idea of 3DGS (Wu et al., 2023; Luiten et al., 2023; Yang et al., 2024; Duan et al., 2024; Sun et al., 2024). Dynamic3DGS (Luiten et al., 2023) optimize the attributes of existing Gaussians to deal with new frames and perform tracking. 4DGS (Wu et al., 2023) build a multi-resolution voxel planet to compute voxel feature with timesteps. Realtime4DGS (Yang et al., 2024) build a 4D Gaussian structure and condition it to 3D Gaussian with a given timestep. 3DGStream (Sun et al., 2024) focuses on online training and builds a transformation cache for optimization. However, despite being an online method, 3DGStream continuously prunes Gaussians during training, making it impossible to perform 3D point tracking across all time frames. While all these methods benefit from the efficiency of differentiable rasterization, they fail to leverage the internal structural information of the real world and still suffer from notable training time.

Our method is mainly inspired by Dynamic3DGS (Luiten et al., 2023) and focuses on the online dynamic scenes (Sun et al., 2024; Li et al., 2022a; Wang et al., 2023; Song et al., 2023), where the method must continually deal with new incoming frames. To make online training much more efficient, we propose a multi-level structure for 3D Gaussians with a new deformation optimization strategy. In addition, our explicit deformation format allows for broad applications like object insertion and part segmentation.

Recent advances in dynamic Gaussian splatting, such as SC-GS (Huang et al., 2024), utilize control points to compress the motion information of Gaussians, transitioning from per-Gaussian training to per-control point training. While SC-GS (Huang et al., 2024) also seeks to optimize Gaussian representations, unlike our method, it focuses on using a single-layer Gaussian structure to improve rendering quality and conducting tasks like scene editing. In contrast, our approach leverages a multi-layer, coarse-to-fine structure to significantly enhance training efficiency. Furthermore, our method is designed for online tasks, while SC-GS is tailored for offline tasks.

**Dynamic Novel-View Synthesis Datasets** for online methods must provide multi-view inputs for each frame. As opposed to offline methods, online methods can only reconstruct one timestep of the scene at a time, with each timestep being initialized using the previous timestep's representation.

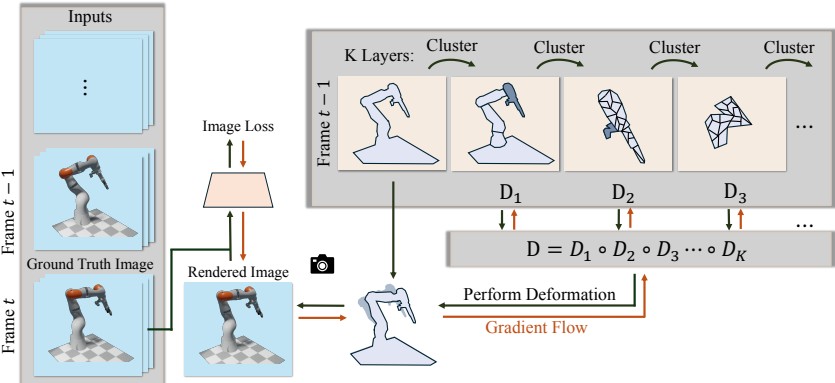

Figure 2: Our method first utilizes the Gaussians from the previous frame $t-1$ and the new inputs for frame $t$ to learn the deformation $D$ between these two frames. These Gaussians are organized into coarse-to-fine multi-layer clusters with $K$ layers. For each cluster layer, we learn a deformation function. Finally, the deformation $D$ of each Gaussian is obtained by nesting these deformation functions.

Therefore, datasets commonly used in offline dynamic synthesis, such as Pumarola et al. (2021) and Park et al. (2021b), cannot be applied in our case. Moreover, our multi-layer, coarse-to-fine design offers a more efficient way to model dynamic Gaussians. It significantly accelerates the convergence during training while preserving the ability to model detailed deformations. Datasets such as Li et al. (2022c) and Broxton et al. (2020), although appearing complex, involve only small-scale movements. As a result, they are not suitable for evaluating our method's capability to model Gaussian dynamics. In the end, we selected accelerated versions of datasets Abou-Chakra et al. (2023) and Luiten et al. (2023) for testing, which meet the aforementioned requirements. For further details, please refer to Sec 4.1.

## 3 METHOD

**Overview**  Our method utilizes online dynamic scene reconstruction, meaning that for each new time frame, we only need the reconstruction result from the previous frame and perform a single deformation prediction. Firstly, we need to perform static scene reconstruction for the initial frame (frame 0) of the entire video. This process follows a standard 3D Gaussian Splatting (Kerbl et al., 2023) procedure. Given multi-view observations of a static scene $(I_{0,1}, I_{0,2}, \ldots, I_{0,N})$ and their corresponding camera poses $(C_1, C_2, \ldots, C_N)$, we need to train a module $\Theta_0$ that contains parameters of all the Gaussians. This enables us to generate a predicted image $\hat{I}$ for any input camera pose $C$, such that $\hat{I} = \Theta_0(C)$.

Based on this, we can proceed with subsequent online dynamic scene reconstruction. To be more specific, we use $S_0, S_1, \ldots, S_T$ to represent the dynamic scene from time frame 0 to time frame $T$. For each time frame $t$, we have a sequence of images $I_{t,1}, I_{t,2}, \ldots, I_{t,N}$ from the cameras. Our goal is to train a representation $\Theta$ that can fit the scenes $S_0, S_1, \ldots, S_T$. Given an arbitrary camera $C$ at time frame $t$, we can predict the image as $\hat{I} = \Theta_t(C)$.

### 3.1 DYNAMIC GAUSSIAN SPLATTING

In this section, we present our method for learning the dynamic scene representation $\Theta$ for the dynamic scene $\{S_0, S_1, \ldots, S_T\}$. In online dynamic scene reconstruction, we only need to predict the deformation of the scene between two frames based on the reconstruction results of the previous frame and the input observations of the current time frame.

Assuming that the Gaussians of frame $t-1$ have been reconstructed, we need to predict the deformation for frame $t$ and obtain the scene representation $\Theta_t$ for it. To be concrete, we want to predict the deformation $D_t$ that satisfies the following equation:

$$\Theta_t = D_t(\Theta_{t-1}). \tag{1}$$

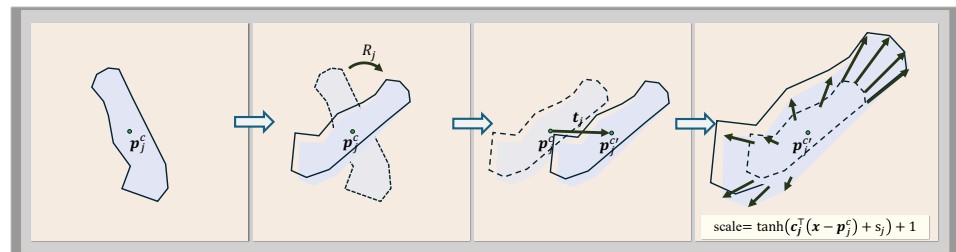

Figure 3: Illustration of the deformation function parameters: Rotation (second), translation (third), and scaling (fourth) applied to a cluster. Initial state (first).

Thus, for any given $t$ and $C$, we have

$$\Theta(t, C) = D_t(D_{t-1}(\cdots D_1(\Theta_0)\cdots))(C). \tag{2}$$

Consider the changes of a single Gaussian $g_i$ in the scene at time frame $t$ during deformation. Recall that its representation is defined as

$$G_{t,i}(\mathbf{X}) = e^{-\frac{1}{2}(\mathbf{X}-\boldsymbol{p}_{t,i})^T \boldsymbol{\Sigma}_{t,i}^{-1}(\mathbf{X}-\boldsymbol{p}_{t,i})}. \tag{3}$$

This is a probability density function of the position $\mathbf{X}$ in which $\boldsymbol{p}_{t,i}$ is the centroid position, and $\boldsymbol{\Sigma}_{t,i}$ is the covariance matrix. In the deformation process $D_t$, we assume that the corresponding deformation function of position $x$ is $\Phi_t$ which satisfies

$$\Phi_t(\boldsymbol{p}_{t-1,i}) = \boldsymbol{p}_{t,i}. \tag{4}$$

If we substitute Eq. (4) into Eq. (3), we can obtain the Gaussian expression at time frame $t$:

$$G_{t,i}(\mathbf{X}) = \exp\left(-\frac{1}{2}(\Phi_t^{-1}(\mathbf{X}) - \boldsymbol{p}_{t-1,i})^T \boldsymbol{\Sigma}_{t-1,i}^{-1}(\Phi_t^{-1}(\mathbf{X}) - \boldsymbol{p}_{t-1,i})\right). \tag{5}$$

If we approximate $\Phi_t$ with a first-order approximation as mentioned in Xie et al. (2024), we can obtain the following,

$$G_{t,i}(\mathbf{X}) = \exp\left(-\frac{1}{2}(\mathbf{X} - \boldsymbol{p}_{t,i})^T (\nabla_{\boldsymbol{p}_{t-1,i}}(\Phi_t)\boldsymbol{\Sigma}_{t-1,i}\nabla_{\boldsymbol{p}_{t-1,i}}(\Phi_t)^T)^{-1}(\mathbf{X} - \boldsymbol{p}_{t,i})\right). \tag{6}$$

Therefore, if we compare Eq. (6) with Eq. (3), we can obtain the deformed centroid position $\boldsymbol{p}_t$ and the covariance matrix $\boldsymbol{\Sigma}_t$ as follows:

$$\begin{aligned}\boldsymbol{p}_{t,i} &= \Phi_t(\boldsymbol{p}_{t-1,i}), \\ \boldsymbol{\Sigma}_{t,i} &= \nabla_{\boldsymbol{p}_{t-1,i}}(\Phi_t)\boldsymbol{\Sigma}_{t-1,i}\nabla_{\boldsymbol{p}_{t-1,i}}(\Phi_t)^T.\end{aligned} \tag{7}$$

This means that if we can learn the deformation function $\Phi_t$ of the scene, we can use Eq. (7) directly to update the parameters of all the Gaussians. Thus, our task is transformed into learning $\Phi_t$, which will be discussed in the following sections.

## 3.2 COARSE-TO-FINE CLUSTERING STRUCTURE

The deformation function $\Phi_t$ can be a complicated non-linear one for the entire scene, making it hard for us to directly learn it. An intuitive idea is that if we can cluster points that are close in space and make an approximation that all the Gaussians within one cluster follow the same deformation function, then the difficulty of learning the deformation function as a whole will be reduced. Also, with this clustering structure, we can make the learning process more efficient than learning it for each Gaussian independently. Which will be revealed in the experiment results. Furthermore, the deformation function within one cluster can be constructed using deformations such as rotation, translation, and scaling, making it possible to parameterize $\Phi_t$ in an explicit form.

The intuition is that one small chunk of the object is nearly rigid, thus its movement can be represented by a transformation and a rotation around its centroid. Also, to increase the flexibility, we can add a scaling factor. The deformation function $\Phi_t$ within one cluster $j$ can be represented as

$$\boldsymbol{x}_d = (R_j(\boldsymbol{x} - \boldsymbol{p}_j^c) + \boldsymbol{t}_j) \cdot (\tanh(\boldsymbol{c}_j^\top(\boldsymbol{x} - \boldsymbol{p}_j^c) + s_j) + 1), \tag{8}$$

where $\boldsymbol{x}$ is the position of the point, $\boldsymbol{x}_d$ is the corresponding position after deformation, $\boldsymbol{p}_j^c$ is the centroid of the cluster, $R_j$ is the rotation matrix (stored as a quaternion to ensure that it represents a rotation), $\boldsymbol{t}_j$ is the translation vector, and $(\tanh(\boldsymbol{c}_j^\top (\boldsymbol{x} - \boldsymbol{p}_j^c) + s_j) + 1)$ as a whole is the scaling factor. This design ensures the scaling factor remains in the range $(0, 2)$, preventing potential NaN problems during training. Additionally, the scaling factor is a flexible, trainable linear function of $(\boldsymbol{x} - \boldsymbol{p}_j^c)$, with $\boldsymbol{c}_j$ and $s_j$ as its parameters, allowing for adaptable scaling within a single cluster. Fig. 3 illustrates the specific meaning of each parameter. In summary, to represent the deformation function $\Phi_t$ within cluster $j$, we need to learn trainable parameters $R_j$, $\boldsymbol{t}_j$, $\boldsymbol{c}_j$ and $s_j$.

The previously discussed content addresses the deformation formulation problem within a single-layer cluster. To provide more flexibility in learning the deformation of the entire scene, we employ a coarse-to-fine multi-layer cluster structure. Initially, we cluster all Gaussians using K-means based on their centroids. Subsequently, by merging neighboring clusters, we acquire a coarser layer of clusters. This process is iteratively repeated until we obtain the coarsest layer of clusters. Specifically, suppose a point $\boldsymbol{p}_{t-1}$ belongs to clusters $j_1$, $j_2$, ..., $j_K$ at each layer respectively ($K$ is the number of layers), and the deformation function within cluster $j_k$ is $\phi_{k,j_k}$. Then, for the point $\boldsymbol{p}_{t-1}$ at the $t-1$-th frame

$$\boldsymbol{p}_t = \phi_{K,j_K}(\cdots(\phi_{2,j_2}(\phi_{1,j_1}(\boldsymbol{p}_{t-1})))\cdots). \qquad (9)$$

In our implementation, $K = 3$. The multi-layer clustering process is shown in Fig. 2.

At the coarsest level, clusters are expected to make broad approximations of the scene's deformation. While this coarse clustering might not always align perfectly with the underlying rigid parts, the purpose is to rapidly bring the Gaussians closer to an optimal solution. Fine-level clusters, operating at higher resolutions, can then start optimization from an improved baseline, requiring fewer iterations to refine the deformation. This hierarchical approach reduces training cost while retaining the ability to express detailed motion.

To further enhance the fine-tuning capability of each Gaussian, we introduce three additional parameters for each Gaussian, which are $\Delta\boldsymbol{p}$, $\Delta R$, and $\Delta\boldsymbol{s}$, corresponding to delta in centroids positions, rotations, and scalings. These delta values are applied to the Gaussians after they have been deformed by the deformation function.

### 3.3 LEARNING THE DEFORMATION

Based on the previously mentioned deformation process, we present our method for learning deformation. Specifically, our method for learning deformations consists of two key stages.

*Initialization Stage:* We begin by training Gaussians on a static scene using observations from the first frame. Following this, we perform a coarse-to-fine multilevel clustering of the Gaussian centroids, which typically needs to be done only once during initialization. However, if there are significant changes in the scene, this clustering can be recomputed mid-training to adapt to the new conditions.

*Training Stage:* Once initialization is complete, we proceed to the training phase, where we optimize deformation parameters frame by frame. For each input frame, we combine the current input images with the Gaussians from the previous frame to predict the deformation. Through backpropagation of 2D loss, we iteratively refine the deformation function. The deformation parameters for each frame are initialized by inheriting those from the previous frame, based on the assumption that the deformations between consecutive frames will be similar.

### 3.4 OPTIMIZATION DETAILS

In this subsection, we introduce some optimization details. In addition to the 2D image losses used in most Gaussian Splatting methods, following Luiten et al. (2023), we also use local-rigidity loss, isometry loss, and rotation loss to restrict the movement of Gaussians in large regions of the same color. Furthermore, we add ratio loss $L^{\text{ratio}}$ and scale loss $L^{\text{scale}}$ to prevent the generation of Gaussians that are excessively large or elongated, helping to reduce artifacts during the deformation process. The explicit forms of these losses are $L^{\text{ratio}} = \frac{1}{N} \sum_{i=1}^{N} \max\left(0, \frac{\max(\text{scale}_{i,t})}{\min(\text{scale}_{i,t})} - \text{max\_ratio}\right)$ and $L^{\text{scale}} = \frac{1}{N} \sum_{i=1}^{N} \max(0, \text{scale}_{i,t} - \text{max\_scale})$, where $\text{scale}_{i,t}$ is the scaling vector of Gaussian $i$ at time $t$, and max\_ratio and max\_scale are hyper-parameters. After the first round of the static stage,

Table 1: Online methods rendering results on the FastParticle and Panoptic datasets. Values represent mean metrics across all testing views. Top-2 methods are bolded.

| Metrics | Method | FastParticle | | | | | | Panoptic | | | | | |
|---|---|---|---|---|---|---|---|---|---|---|---|---|---|
| | | Robot | Spring | Wheel | Pendulums | Robot-Task | Cloth | Basketball | Boxes | Football | Juggle | Softball | Tennis |
| PSNR↑ | Ours$_{100}$ | **29.46** | **30.28** | **27.95** | **30.6** | 27.67 | **31.68** | **30.25** | **29.46** | **30.47** | **31.12** | **31.02** | **30.21** |
| | Dynamic3DGS$_{100}$ (Luiten et al., 2023) | 21.28 | 23.66 | 24.14 | 24.98 | 23.41 | 21.44 | 29.48 | 29.20 | 30.05 | 30.96 | 30.64 | 29.77 |
| | Dynamic3DGS$_{2000}$ (Luiten et al., 2023) | **30.23** | **30.88** | **28.59** | **31.23** | 29.36 | **32.91** | **30.01** | **29.29** | **30.4** | **31.04** | **30.88** | **30.11** |
| SSIM↑ | Ours$_{100}$ | **0.96** | **0.97** | **0.94** | **0.97** | **0.95** | **0.97** | **0.93** | **0.93** | **0.94** | **0.94** | **0.94** | **0.94** |
| | Dynamic3DGS$_{100}$ (Luiten et al., 2023) | 0.90 | 0.93 | 0.89 | 0.94 | 0.92 | 0.92 | 0.92 | 0.93 | 0.93 | 0.94 | 0.94 | 0.94 |
| | Dynamic3DGS$_{2000}$ (Luiten et al., 2023) | **0.97** | **0.97** | **0.94** | **0.97** | **0.97** | **0.98** | **0.92** | **0.93** | **0.93** | **0.94** | **0.94** | **0.94** |
| LPIPS↓ | Ours$_{100}$ | **0.09** | **0.04** | **0.07** | **0.06** | **0.10** | **0.06** | **0.21** | **0.20** | **0.20** | **0.20** | **0.20** | **0.19** |
| | Dynamic3DGS$_{100}$ (Luiten et al., 2023) | 0.15 | 0.08 | 0.11 | 0.09 | 0.13 | 0.11 | 0.22 | 0.21 | 0.21 | 0.20 | 0.21 | 0.21 |
| | Dynamic3DGS$_{2000}$ (Luiten et al., 2023) | **0.08** | **0.04** | **0.06** | **0.05** | **0.09** | **0.05** | **0.22** | **0.21** | **0.21** | **0.21** | **0.21** | **0.21** |

Table 2: General dynamic methods rendering results on the FastParticle and Panoptic datasets. Values represent mean metrics across all testing views. The best method is bolded.

| Metrics | Method | FastParticle | | | | | | Panoptic | | | | | |
|---|---|---|---|---|---|---|---|---|---|---|---|---|---|
| | | Robot | Spring | Wheel | Pendulums | Robot-Task | Cloth | Basketball | Boxes | Football | Juggle | Softball | Tennis |
| PSNR↑ | Ours | **29.46** | **30.28** | **27.95** | **30.60** | 27.67 | **31.68** | **30.25** | **29.46** | **30.47** | **31.12** | **31.02** | **30.21** |
| | Dynamic3DGS (Luiten et al., 2023) | 21.28 | 23.66 | 24.14 | 24.98 | 23.41 | 21.44 | 29.48 | 29.2 | 30.05 | 30.96 | 30.64 | 29.77 |
| | RealTime4DGS (Yang et al., 2024) | 25.97 | 22.54 | 23.86 | 26.25 | 24.72 | 22.16 | 25.51 | 27.59 | 26.48 | 27.63 | 26.73 | 27.09 |
| | 4DGS (Wu et al., 2023) | 25.86 | 24.93 | 26.56 | 27.35 | **28.00** | 27.89 | 23.26 | 28.02 | 27.04 | 28.10 | 26.01 | 27.54 |
| | SC-GS(no-pretraining) (Huang et al., 2024) | 15.76 | 17.08 | 16.89 | 17.90 | 16.42 | 14.58 | 19.72 | 21.43 | 20.66 | 20.87 | 21.03 | 21.10 |
| | SC-GS(pretraining) (Huang et al., 2024) | 22.31 | 25.60 | 24.10 | 27.32 | 26.49 | 26.95 | 19.42 | 21.02 | 20.17 | 20.62 | 21.11 | 21.02 |
| SSIM↑ | Ours | **0.96** | **0.97** | **0.94** | **0.97** | **0.95** | **0.97** | **0.93** | 0.93 | **0.94** | **0.94** | **0.94** | **0.94** |
| | Dynamic3DGS (Luiten et al., 2023) | 0.90 | 0.93 | 0.89 | 0.94 | 0.92 | 0.92 | 0.92 | **0.93** | 0.93 | **0.94** | **0.94** | **0.94** |
| | RealTime4DGS (Yang et al., 2024) | 0.93 | 0.91 | 0.89 | 0.93 | 0.93 | 0.91 | 0.89 | 0.92 | 0.91 | 0.92 | 0.91 | 0.92 |
| | 4DGS (Wu et al., 2023) | 0.93 | 0.93 | 0.91 | 0.94 | **0.95** | 0.95 | 0.87 | 0.92 | 0.91 | 0.92 | 0.91 | 0.92 |
| | SC-GS(no-pretraining) (Huang et al., 2024) | 0.75 | 0.78 | 0.80 | 0.66 | 0.76 | 0.73 | 0.69 | 0.70 | 0.69 | 0.71 | 0.72 | 0.70 |
| | SC-GS(pretraining) (Huang et al., 2024) | 0.90 | 0.95 | 0.87 | 0.95 | 0.94 | 0.94 | 0.68 | 0.69 | 0.68 | 0.71 | 0.71 | 0.70 |
| LPIPS↓ | Ours | **0.09** | **0.04** | **0.07** | 0.06 | **0.10** | **0.06** | **0.21** | **0.20** | **0.20** | **0.20** | **0.20** | **0.19** |
| | Dynamic3DGS (Luiten et al., 2023) | 0.15 | 0.08 | 0.11 | 0.09 | 0.13 | 0.11 | 0.22 | 0.21 | 0.21 | **0.20** | 0.21 | 0.21 |
| | RealTime4DGS (Yang et al., 2024) | 0.13 | 0.11 | 0.12 | 0.10 | 0.13 | 0.13 | 0.26 | 0.22 | 0.23 | 0.22 | 0.23 | 0.23 |
| | 4DGS (Wu et al., 2023) | 0.12 | 0.08 | 0.10 | 0.09 | 0.11 | 0.09 | 0.32 | 0.25 | 0.27 | 0.25 | 0.26 | 0.25 |
| | SC-GS(no-pretraining) (Huang et al., 2024) | 0.27 | 0.17 | 0.15 | 0.21 | 0.26 | 0.25 | 0.44 | 0.44 | 0.43 | 0.42 | 0.41 | 0.42 |
| | SC-GS(pretraining) (Huang et al., 2024) | 0.12 | **0.04** | 0.12 | **0.04** | **0.07** | 0.07 | 0.45 | 0.43 | 0.44 | 0.41 | 0.41 | 0.42 |

we fix the opacity and background logit of the Gaussians. For better rendering results, we make the color trainable, allowing it to better adapt to different lighting conditions. Specifically, in terms of appearance modeling, we follow the approach of Dynamic3DGS (Luiten et al., 2023), assigning each Gaussian a trainable 3D RGB vector instead of using spherical harmonics (SH). Additionally, we add a soft RGB loss to constrain the changes in color. Regarding the coarse-to-fine clustering, in our experiments, we use a structure with $K = 3$, where the clusters in the coarser layer are obtained by merging clusters from the finer layer. The finest layer clusters are obtained using KMeans of Gaussian centroid positions. The merging method involves calculating the average centroid position of Gaussians in each cluster, and then performing Agglomerative Clustering based on this. The final numbers of clusters at each layer are $64, 320, 1280$.

## 4 EXPERIMENTS

### 4.1 DATASET PREPARATION

We conduct our experiments on two datasets: the Panoptic dataset (Luiten et al., 2023), which includes six real-world dynamic scenes (Basketball, Boxes, Football, Juggle, Softball, and Tennis), and the synthetic FastParticle dataset (Abou-Chakra et al., 2023), containing six highly dynamic scenes (Robot, Spring, Wheel, Pendulums, Robot-Task, and Cloth). As mentioned in Sec. 2, we deliberately chose these datasets with challenging motion patterns to evaluate our method's ability to quickly converge Gaussians in a very short training period. The large motion between frames in these datasets increases the difficulty of rapid convergence, making them ideal for testing the robustness of our approach. To further amplify this challenge, we accelerated the motion in the FastParticle dataset. Additional details are available in the appendix.

### 4.2 COMPARISONS

In this section, we compare our method with the state-of-the-art dynamic Gaussian Splatting methods on View-Synthesis tasks. These methods include Dynamic3DGS (Luiten et al., 2023), Real-Time4DGS (Yang et al., 2024), 4DGS (Wu et al., 2023) and SC-GS (Huang et al., 2024). Among them, Dynamic3DGS (Luiten et al., 2023) adopts the same online dynamic scene reconstruction approach as ours, while the other two are offline methods.

For evaluation metrics, we use the PSNR, SSIM, and LPIPS (Wang et al., 2004; Zhang et al., 2018). In the experiments, since training speed is greatly influenced by implementation and hardware, for a fair comparison, it is most reasonable to compare the rendering results at the same iteration number. To eliminate concerns about runtime speed. On our single NVIDIA A40 GPU, training 100 iterations takes 1-3 seconds.

Table 3: 2D tracking results on the FastParticle and Panoptic datasets. Values represent mean metrics across all testing trajectories. The best method is bolded.

| Metrics | Method | FastParticle | | | | | | Panoptic | | | | | |
|---|---|---|---|---|---|---|---|---|---|---|---|---|---|
| | | Robot | Spring | Wheel | Pendulums | Robot-Task | Cloth | Basketball | Boxes | Football | Juggle | Softball | Tennis |
| 2D MTE↓ | Ours$_{100}$ | **0.80%** | **0.17%** | **14.88%** | **0.50%** | **0.82%** | **0.24%** | **0.57%** | **0.22%** | **7.64%** | **8.15%** | **0.39%** | **1.72%** |
| | Dynamic3DGS$_{100}$ (Luiten et al., 2023) | 7.84% | 2.26% | 18.53% | 3.51% | 4.42% | 2.33% | 15.85% | 4.95% | 9.29% | 12.42% | 16.43% | 25.19% |

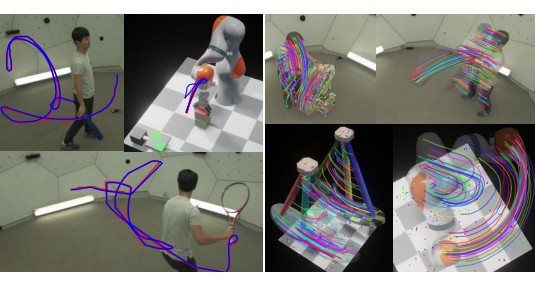

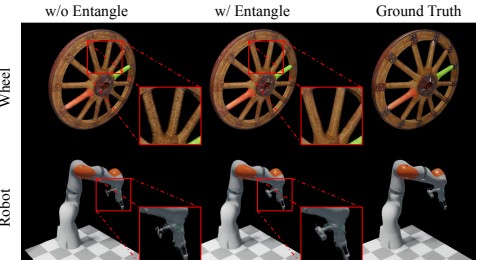

Figure 4: Left: Comparing our tracking result (blue) to the ground truth (red). Right: Visualization of our tracking results.

Figure 5: Ablation study of the entangled covariance matrix.

For online dynamic methods, we conducted experiments comparing our method with Dynamic3DGS (Luiten et al., 2023) on both the FastParticle and Panoptic datasets. The results are shown in Table 1. Since both methods follow the paradigm of first training a static scene and then performing Gaussian Splatting training frame by frame, we fixed the number of training iterations between every two frames to 100 and 2000 for comparison. Here, we provide the same static checkpoints for both methods for fairness. As mentioned earlier, our method significantly reduces the number of iterations required to achieve satisfactory rendering results. From the results in the tables, it can be seen that our method achieves results comparable to Dynamic3DGS (Luiten et al., 2023) at 2000 iterations with only 100 iterations of training between frames, and it far surpasses Dynamic3DGS (Luiten et al., 2023) at 100 iterations. Fig. 6 shows the rendering results of both methods at 100 iterations, qualitatively demonstrating that our method can converge and achieve satisfactory visual results after being trained for only 100 iterations per time frame.

For general dynamic methods (both online and offline), we compared our method with the other four methods on both datasets. For fairness, the two online methods trained 100 iterations per frame, while for the offline methods, we set their total iterations to 100 multiplied by the total number of frames. Similarly, we provided all methods with the same static scene checkpoints for fair comparisons. SC-GS (Huang et al., 2024) is a special case because it involves two stages: the first stage requires 10,000 iterations solely to establish control points, and the second stage begins the actual rendering training. Therefore, we provide two metrics: pretraining refers to the scenario where SC-GS undergoes 10,000 iterations to establish control points before continuing with the same number of iterations as our method, effectively adding 10,000 extra iterations. No-pretraining refers to the case where we skip the additional 10,000 iterations and directly start the rendering training. The results are shown in Table 2. As can be seen from the table, our method achieved the best results across both datasets.

Additionally, we evaluated our method's point-tracking capability. Due to the challenge of obtaining 3D ground-truth tracking labels, we manually annotated keypoints for all frames from a selected camera view in each scene, using these as ground-truth data. Details of the annotation process are provided in the appendix. For tracking, we projected all Gaussian centroids in each frame onto the camera plane to obtain predicted 2D points. We then selected candidate points within 10 pixels of the ground-truth 2D keypoint from the first frame, choosing the one with the highest metric value as the final tracked point. This step was necessary because the candidates corresponded to different depths, and the 2D ground-truth coordinates alone were insufficient for determining which point to track. The candidate with the best metric was considered the 3D-consistently aligned point. We used the 2D Median Trajectory Error (MTE) as the metric, following Dynamic3DGS (Luiten et al., 2023). In Table 3, we report the normalized MTE, which is the pixel error normalized by the image diagonal length, along with visualizations of the tracking results in Fig. 4. We compare our method against Dynamic3DGS (Luiten et al., 2023), selected for its superior rendering performance and as the only baseline aligning with our settings. Our tracking outcomes significantly outperform the baseline across all scenes with the same number of training iterations. Notably, the "Wheel" scene

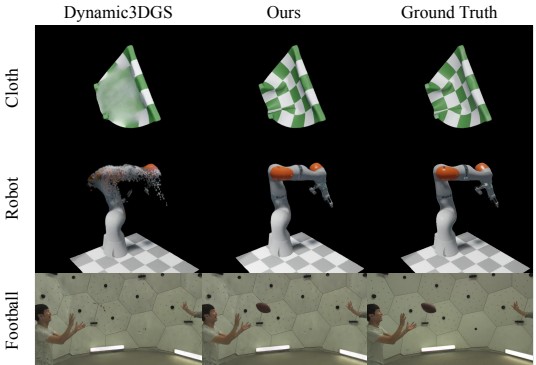
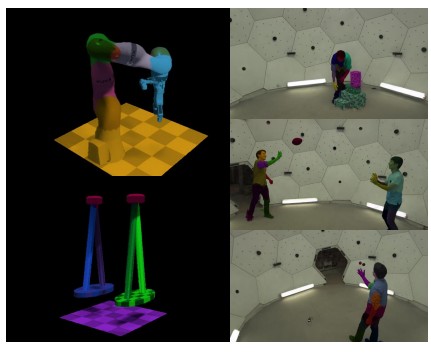

Figure 6: Visual comparison of rendering results on FastParticle after 100 iterations per frame training.

Figure 7: Articulated objects segmentation results.

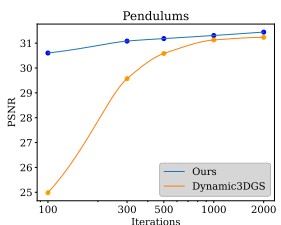
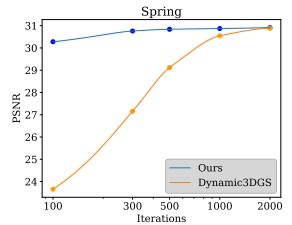
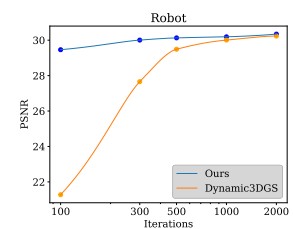

Figure 8: Convergence speed comparison between our method and Dynamic3DGS (Luiten et al., 2023) on the FastParticle dataset. The x-axis shows the number of training iterations per frame, and the y-axis represents the mean PSNR across all testing views.

exhibits a high 2D MTE due to the object's strong symmetry, leading to ambiguity in its rotational trajectory (see the appendix for the scene image).

## 4.3 ABLATION STUDY

In this section, we conduct ablation studies to analyze the effectiveness of our method. We first analyze the convergence speed of our method and compare it with Dynamic3DGS (Luiten et al., 2023). Then, we study the influence of the number of clustering layers on the rendering results. Finally, we analyze the effect of the entangled covariance matrix on the rendering results.

**Analyse of training iterations**  To compare the convergence speed of our method and Dynamic3DGS (Luiten et al., 2023), we trained both methods for different iterations and evaluated their rendering results at these iterations. We trained both methods on the FastParticle dataset. We show the results in Fig. 8, where the x-axis represents the number of training iterations between every two time frames, and the y-axis represents the mean PSNR among all of the testing views. It can be observed that our method converges much faster than Dynamic3DGS (Luiten et al., 2023), consistently outperforming Dynamic3DGS (Luiten et al., 2023) at the same number of iterations. After 2000 iterations, both methods converge at the same point, which also confirms that our method is very close to convergence after training for just 100 iterations.

**Number of Cluster Layers**  In our multi-layer clustering design, we choose the number of layers $K$ to be 3. Here, we conduct experiments to analyze the influence of $K$ on the rendering results, and also to validate the effectiveness of our multi-layer clustering design. We conduct our experiments on the FastParticle dataset, and the results are shown in Table 4. It can be seen that the results of our method with $K = 3$ are much better than those with $K = 1$ across all metrics and scenes, revealing that the coarse-to-fine structure can significantly reduce the number of training iterations, validating the intuition that moving large clusters of Gaussians at once can more quickly find suitable positions, thereby reducing unnecessary adjustments of Gaussian positions.

**Entangled Covariance Matrix**  As shown in Eq. (7), in our method, our Gaussians' covariance matrixes are not separately learned. Instead, they are coupled with the deformation of centroid positions. This makes it easier for our method to learn the correct rotations and scaling of the

| Metrics | Method | Particle | | | | | |
|---|---|---|---|---|---|---|---|
| | | Robot | Spring | Wheel | Pendulums | Robot-Task | Cloth |
| PSNR↑ | Ours, K=3 | **29.46** | **30.28** | **27.95** | **30.60** | **27.67** | **31.68** |
| | Ours, K=1 | 24.56 | 26.16 | 25.12 | 25.94 | 24.73 | 27.77 |
| SSIM↑ | Ours, K=3 | **0.96** | **0.97** | **0.94** | **0.97** | **0.95** | **0.97** |
| | Ours, K=1 | 0.94 | 0.95 | 0.91 | 0.95 | 0.94 | 0.96 |
| LPIPS↓ | Ours, K=3 | **0.09** | **0.04** | **0.07** | **0.06** | **0.10** | **0.06** |
| | Ours, K=1 | 0.12 | 0.07 | 0.10 | 0.08 | 0.12 | 0.07 |

Table 4: Ablation study for the number of cluster layers.

Gaussians. In Fig. 5, we show a comparison between learning the covariance matrix separately from positions and our full implementation.

Using the wheel as an example, it should rotate around its own center. As shown in the left, without coupling, although the positions of the Gaussians are learned correctly, the Gaussians themselves do not rotate accordingly with the wheel, resulting in suboptimal final rendering. In our full implementation, as long as the deformation function is learned correctly, the rotations and scalings of the Gaussians are naturally adjusted accordingly, preventing artifacts where the Gaussians are incorrectly oriented.

## 5 APPLICATIONS: SEGMENT ARTICULATED OBJECTS

Next, we demonstrate another application of the learned deformation information: performing segmentation on articulated objects without any semantic knowledge. Many objects in daily life, although not rigid as a whole, are composed of many rigid parts. As humans, we can distinguish these parts by watching a dynamic video and observing their motions. In this section, we show how to perform segmentation of different parts of an object in a zero-shot manner by only utilizing deformation information.

After training, we can obtain the centroid positions and rotations of Gaussians at each time frame. We then use KMeans clustering to group Gaussians into different clusters based on this information. Specifically, for a given Gaussian $i$, we use the notations $\boldsymbol{p}_{i,t}$ and $R_{i,t}$ to represent its position and rotation matrix at time frame $t$, respectively. The KMeans feature for each Gaussian is a tensor of shape $[T, 15]$, where $T$ is the total number of time frames. This tensor is the concatenation of $\boldsymbol{p}_{i,t}$, flattened $R_{i,t}$, and $\boldsymbol{p}_{i,0}$ repeated $T$ times. Additionally, we multiply three hyperparameters: $\lambda_{\boldsymbol{p}}$, $\lambda_R$, and $\lambda_{\boldsymbol{p}_0}$ to these three parts before concatenation, respectively, to balance their importance.

The intuition behind the KMeans design is that, (1) Gaussians belong to the same part of the object should be close to each other at all times, and (2) the rotations of Gaussians within the same rigid part should be the same.

As illustrated in Fig. 7, we present our segmentation results on the Panoptic and FastParticle datasets. To enhance visualization, we assign different colors to Gaussians belonging to distinct categories before rendering the final outcomes. Notably, our simple K-means algorithm yields highly intuitive segmentation results, regardless of whether the scene comprises synthetic objects (left) or intricate real-world environments (right). This observation serves as indirect evidence that the deformation information captured by our learned Gaussians closely aligns with the actual motion of objects in dynamic scenes.

## 6 CONCLUSION

In this paper, we show that a natural multi-layer structured 3D Gaussian can greatly improve the training speed in dynamic scene rendering. Based on this, a trainable deformation function for multi-level clusters is proposed to achieve high-fidelity rendering results. With these strategies, our LayeredGS can perform very efficient per-frame training with only 1/20 iterations of the state-of-the-art (Luiten et al., 2023), maintaining comparable rendering performance. Compared with previous methods, our LayeredGS explicitly models the deformation and allows applications like articulated object segmentation. As we use the standard 3D Gaussian format, experiments with other 3DGS variants can be future directions.

## 7 ETHICS STATEMENT

As an efficient and accurate dynamic novel-view synthesis method, LayeredGS has the potential to have significant positive social impacts. Specifically, efficient dynamic rendering can make immersive experiences more accessible to a broader audience, potentially enhancing education, training, and entertainment. Due to its capacity for online training, it can also help with the digital twin model in the industry, facilitating remote collaboration and communication. In addition, layeredGS can be applied in robotics and autonomous systems, helping in perception, manipulation, and decision-making.

## 8 REPRODUCIBILITY STATEMENT

To ensure reproducibility, we will open-source the entire project, including the code and the datasets used in our experiments. The code will be made available after publication, and we will provide all the necessary steps to reproduce the results presented in the paper. Additionally, in Sec. A of the appendix, we provide a detailed description of the process for constructing our datasets, ensuring that other researchers can replicate the data preparation pipeline.

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

# LAYEREDGS: EFFICIENT DYNAMIC SCENE RENDERING AND POINT TRACKING WITH MULTI-LAYER DEFORMABLE GAUSSIAN SPLATTING SUPPLEMENTARY MATERIAL

**Anonymous authors**

In this supplementary material, we offer additional details regarding our FastParticle and Panoptic datasets, which provide the necessary context for our experiments. We delve into our method for articulated objects segmentation, presenting full qualitative results that demonstrate its effectiveness across various scenarios. Additionally, we clarify our rationale for maintaining the same number of iterations in our comparisons and present a comparison under equal wall-clock time, showing that our method still outperforms Dynamic3DGS Luiten et al. (2023). We also include visualizations illustrating the multi-layer clustering structure we employ, as well as the manually annotated tracking labels used for evaluating 2D tracking results. Furthermore, we discuss our approach to learning the deformation, emphasizing the two-phase training strategy. Finally, we reflect on limitations, identifying potential areas for future improvement.

## A FASTPARTICLE AND PANOPTIC DATASETS

In this section, we introduce the FastParticle and Panoptic datasets used in our experiments in details. The real-world Panoptic dataset includes six scenes: Basketball, Boxes, Football, Juggle, Softball, and Tennis. Each frame in these scenes comes with segmentation provided by the original authors. Following Luiten et al. (2023), we distinguish between foreground and background in these scenes and utilize background loss and floor loss accordingly. Each scene in this dataset contains 150 frames captured by a total of 31 cameras, with 27 cameras used for training and 4 for testing.

The synthetic FastParticle dataset, which we have accelerated, contains six dynamic scenes: Robot, Spring, Wheel, Pendulums, Robot-Task, and Cloth. After acceleration, these scenes respectively have 35, 18, 38, 24, 35, and 35 frames. As illustrated in fig. I, we show the dynamic evolution of some scenes, highlighting the significant changes between frames. This dataset includes 40 cameras in total, from which we randomly select 4 as testing cameras and the remaining 36 as training cameras.

For all experiments, we provide the same static checkpoints to all baselines. For the 12 scenes across the two datasets, we train for 20,000 iterations to obtain the checkpoints. Due to the varying complexity of the static scenes, 3,000 iterations are sufficient for most FastParticle scenes.

## B ARTICULATED OBJECTS SEGMENTATION

As mentioned in Sec. 5.1. The intuition behind the KMeans design is that, (1) Gaussians belong to the same part of the object should be close to each other at all time, and (2) the rotations of Gaussians within the same rigid part should be the same. The first one can be trivial, here we provide more explanations about the second point. As shown in fig. II, suppose we have a rigid body with its centroid denoted as $C_0$. This rigid body can be considered as a combination of two smaller rigid bodies, with their centroids denoted as $C_1$ and $C_2$, respectively. After rotation, $C_1$ and $C_2$ move to $C_1'$ and $C_2'$. Taking $C_0$ as the origin of the coordinate system, the movement of the rigid body can only be a rotation $R$ around $C_0$, and the two smaller rigid bodies move accordingly. When considering the left smaller rigid body alone, its motion should consist of a translation of its centroid $C_1$ and a rotation $R_1$ around $C_1$. We aim to prove that $R_1 = R$. Therefore, consider a point $P$ on the left rigid body, which moves to point $P'$ after the movement. From the perspective of $C_0$, we

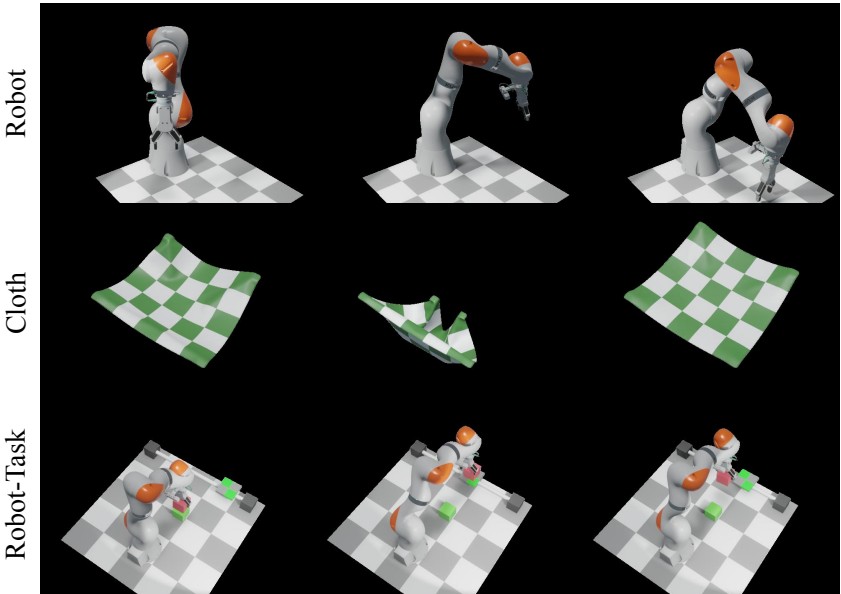

Figure I: This figure shows the evolution of three scenes from the FastParticle dataset, demonstrating the high dynamic characteristics of the accelerated dataset.

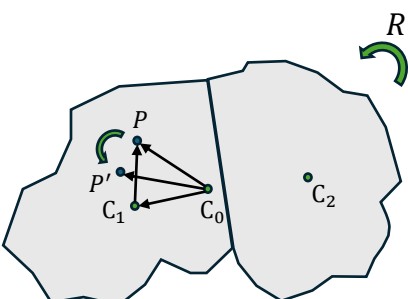

Figure II: Illustration of a rigid body rotating $R$ around its centroid. When considering the rigid body as composed of two smaller rigid bodies, it can be shown that the rotation of each smaller rigid body around its own centroid is the same with $R$.

have

$$\overrightarrow{C_0P'} = R\,\overrightarrow{C_0P}. \tag{1}$$

Also, from the perspective of $C_1$, we can have

$$\begin{aligned}
\overrightarrow{C_0P'} &= R_1\overrightarrow{C_1P} + \overrightarrow{C_0C_1'} \\
&= R_1\overrightarrow{C_1P} + R\,\overrightarrow{C_0C_1}.
\end{aligned} \tag{2}$$

Therefore, we have

$$R\,\overrightarrow{C_1P} = R_1\,\overrightarrow{C_1P}. \tag{3}$$

Since the choice of $P$ is arbitrary, we can conclude that $R_1 = R$. Similarly, we can prove that the rotation of the smaller rigid body on the right is also $R$. The above demonstrates the case where the rigid body is divided into two parts. This conclusion can be generalized to any case of multiple divisions, meaning that all parts of the same rigid body have the same rotation. Returning to our problem, since the rotation of Gaussians is around their centroids, the Gaussians belonging to the same rigid body should have the same rotation.

## C    FULL QUALITATIVE RESULTS

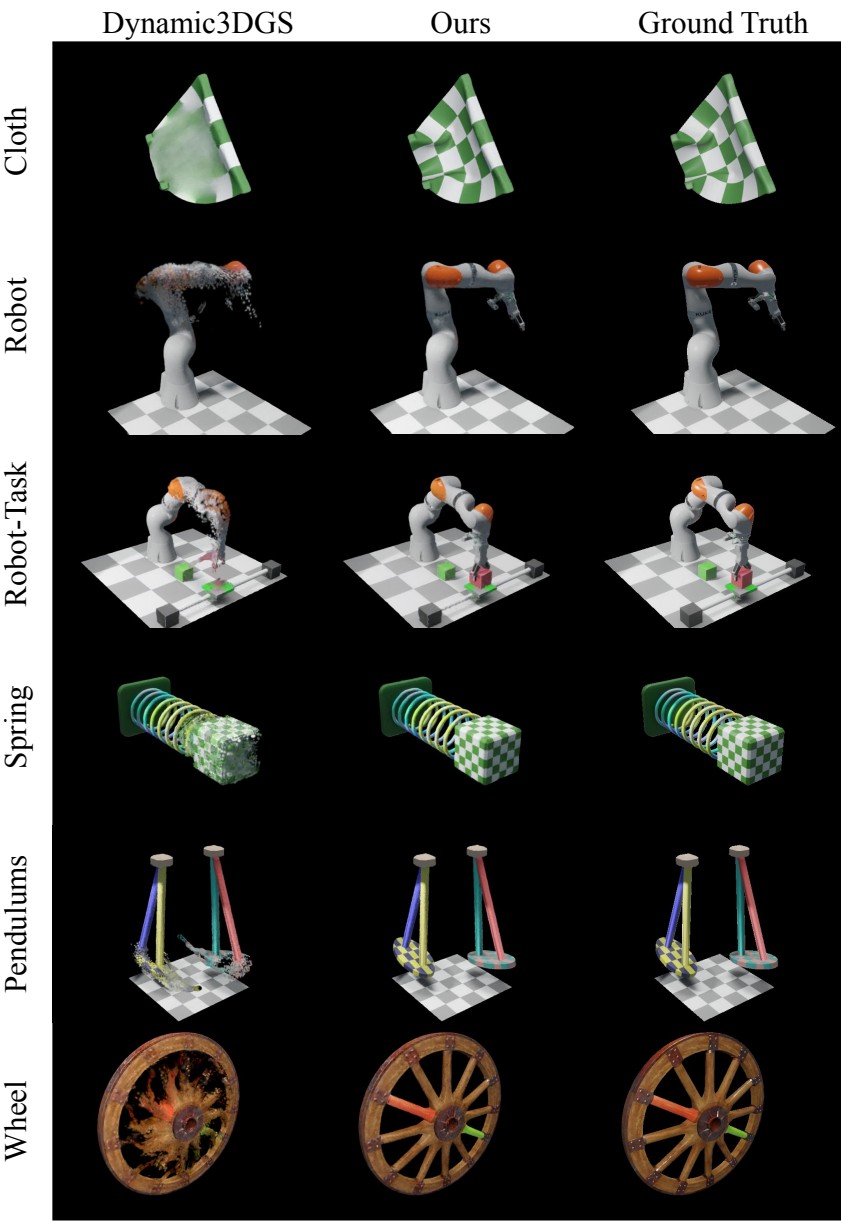

Figure III: Qualitative results on FastParticle

In this section, we provide qualitative results on all 12 scenes from the two datasets. As shown in fig. III and fig. IV, both our method and Luiten et al. (2023) are trained 100 iterations between two consecutive frames.

## D    SAME WALL-CLOCK TIME COMPARISONS

In our experiments, we use the same number of iterations across different methods for consistency. While wall-clock time may vary depending on the specific implementation (e.g., whether CUDA acceleration is employed), the number of iterations reflects the convergence speed of the algorithms. A lower number of iterations indicates faster convergence, showing that the optimization problem is

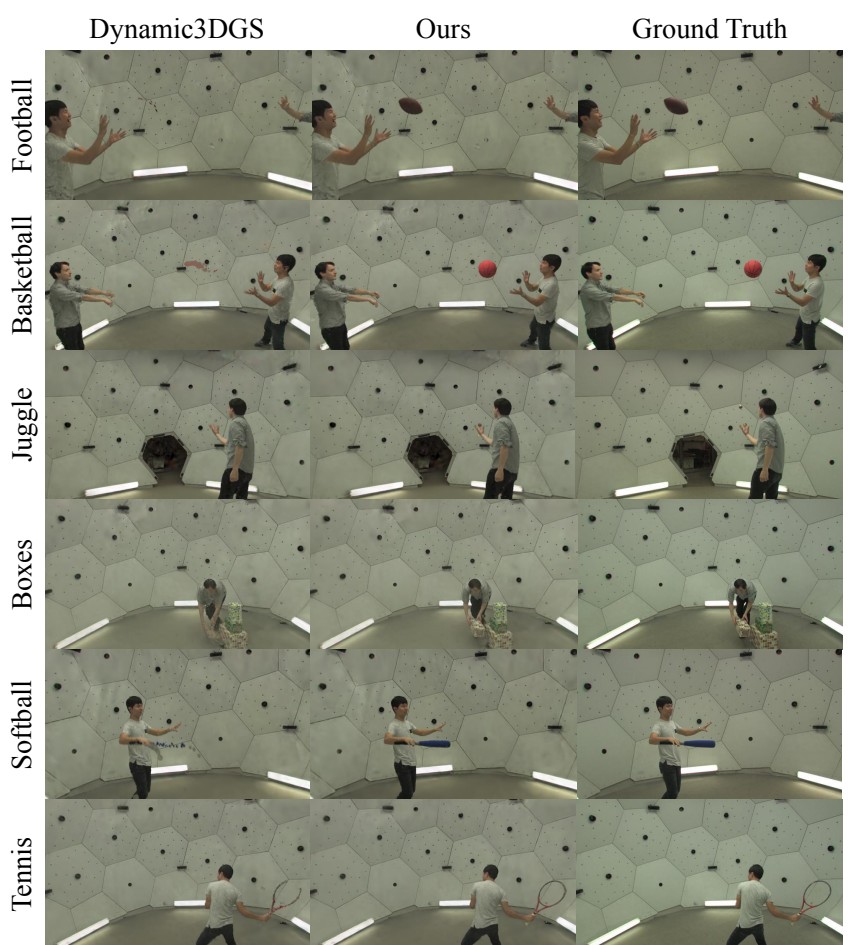

Figure IV: Qualitative results on Panoptic

| Metrics | Method | FastParticle | | | | | |
|---------|--------|-------|--------|-------|-----------|------------|-------|
| | | Robot | Spring | Wheel | Pendulums | Robot-Task | Cloth |
| PSNR↑ | Ours$_{100}$ | **29.46** | **30.28** | **27.95** | **30.60** | **27.67** | **31.68** |
| | Dynamic3DGS$_{300}$ Luiten et al. (2023) | 27.66 | 27.16 | 26.67 | 29.57 | 26.79 | 30.41 |
| SSIM↑ | Ours$_{100}$ | **0.96** | **0.97** | **0.94** | **0.97** | **0.95** | **0.97** |
| | Dynamic3DGS$_{300}$ Luiten et al. (2023) | 0.95 | 0.95 | 0.93 | 0.96 | **0.95** | **0.97** |
| LPIPS↓ | Ours$_{100}$ | **0.09** | **0.04** | **0.07** | **0.06** | **0.10** | **0.06** |
| | Dynamic3DGS$_{300}$ Luiten et al. (2023) | 0.10 | 0.06 | 0.08 | **0.06** | **0.10** | 0.07 |

Table I: Comparison of our method trained with 100 iterations per time frame against Dynamic3DGS.

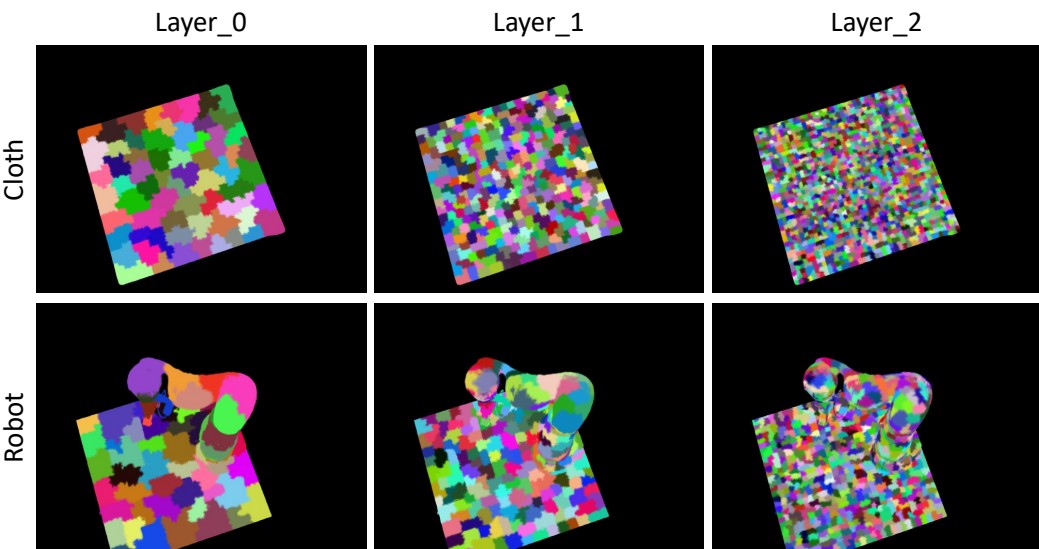

Figure V: Coarse-to-fine multi-layer clustering structures for two objects in the FastParticle dataset.

easier to solve. This practice is commonly used in the evaluation of online methods, as demonstrated in the Dynamic3DGS Luiten et al. (2023) comparison (see Table 1 in their paper), where different methods are also compared using the same number of iterations.

Even when comparing with equivalent wall time, our method remains superior. To further illustrate this, we provide a comparison of our method trained for 100 iterations per frame versus Dynamic3DGS Luiten et al. (2023) trained for 300 iterations per frame on the FastParticle dataset. The results show that our method has an average training speed per iteration approximately twice as fast as Dynamic3DGS Luiten et al. (2023). As seen in table I, despite the difference in iteration count, our method still outperforms Dynamic3DGS Luiten et al. (2023) in terms of both efficiency and final performance.

## E   ILLUSTRATION OF THE MULTI-LAYER STRUCTURE

In fig. V, we show the coarse-to-fine multi-layer clustering structures for two objects in the FastParticle dataset. Different colors in the figure represent different clusters, and for clarification, the same color in different layers does not indicate any correlation between the clusters.

## F   TRACKING LABELS

Here, as shown in fig. VI, we present all manually annotated 2D tracking ground truths. Since the human eye can only track points with distinct features across multiple frames, we only selected such points for annotation.

## G   LEARNING THE DEFORMATION

algorithm 1 summarizes our training process. Initially, we train our Gaussians on the static scene using observations from the first frame. Subsequently, we perform multilevel coarse-to-fine clustering for the centroids of the Gaussians. For each input in every time frame, we use an optimization approach to backpropagate loss and subsequently update our deformation functions.

For potential negative impacts, since LayeredGS can learn deformation information and be used for creating new motions or inserting objects, such applications can be used for fake news to convince people by multi-view renderings. More censorship needs to be established in such cases.

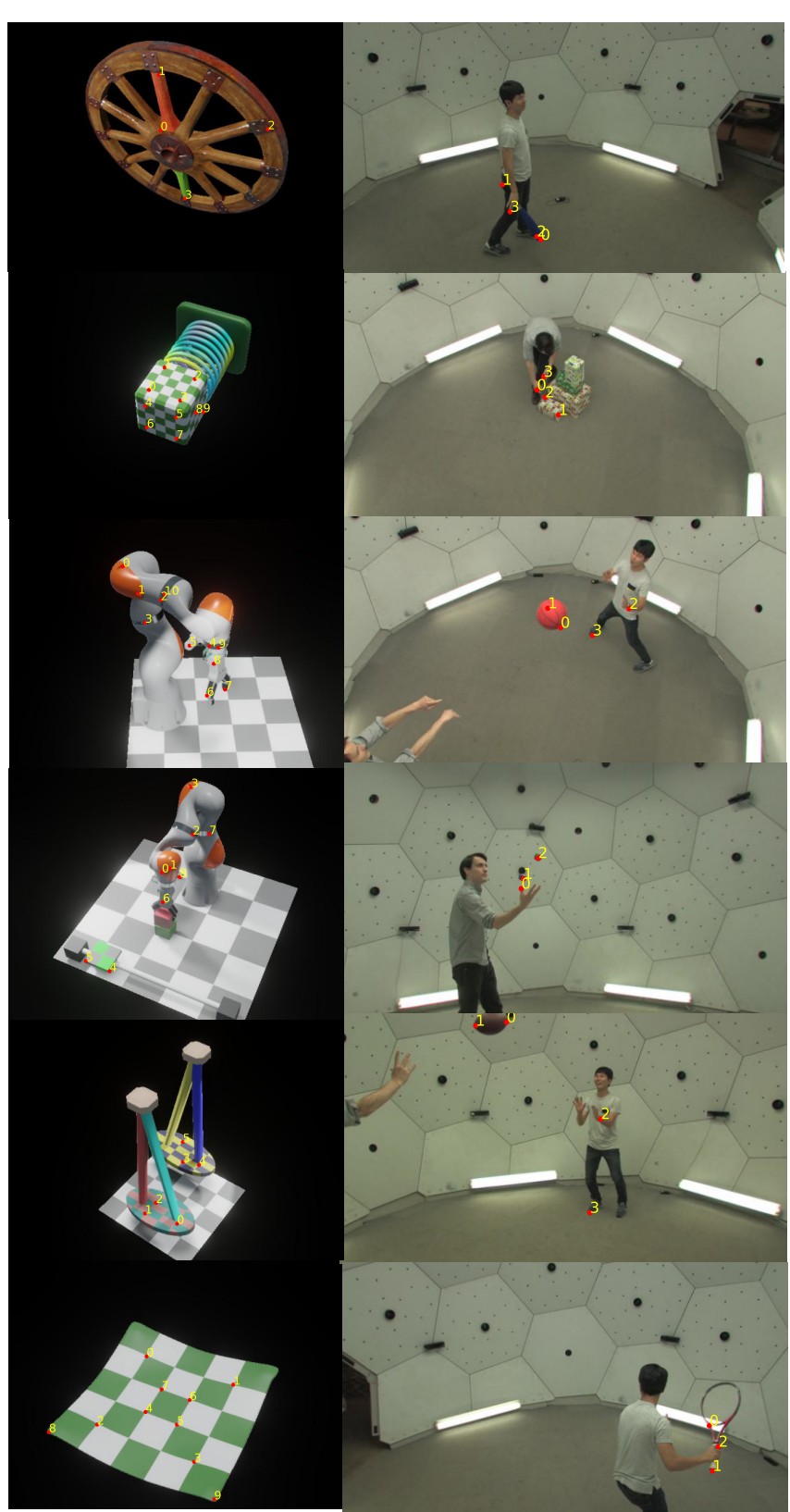

Figure VI: Illustration of our manually annotated tracking ground truths.

**Algorithm 1:** Deformation-based Dynamic Scene Reconstruction Algorithm

**Input:** Images from all frames
$\Theta_{prev} \leftarrow$ Initialization stage (Static Gaussian Splatting);
Do Clustering;
**for** $t$ **in** *time_frames* **do**
    Initialize the Deformation $D$;
    **for** *iter* **in** *max_iters* **do**
        $\Theta_{curr} \leftarrow D(\Theta_{prev})$;
        Images $\leftarrow$ Render($\Theta_{curr}$);
        loss $\leftarrow$ Loss(gt_Images, Images);
        Backpropagate(loss);
    **end**
**end**

## H   LIMITATIONS

While our method significantly reduces training iterations to 100 per frame, achieving real-time training and rendering remains a challenge. Additionally, the learned deformation information is not fully utilized, and the presented articulated object segmentation results are not well refined. Future work will focus on addressing these limitations by exploring real-time training approaches, refining deformation utilization techniques, and developing more sophisticated segmentation methods.

## REFERENCES

Jonathon Luiten, Georgios Kopanas, Bastian Leibe, and Deva Ramanan. Dynamic 3d gaussians: Tracking by persistent dynamic view synthesis. *arXiv preprint arXiv:2308.09713*, 2023.

