# OpenReview forum: "LayeredGS: Efficient Dynamic Scene Rendering and Point Tracking with Multi-Layer Deformable Gaussian Splatting"
_ICLR.cc/2025/Conference — ICLR 2025 Conference Withdrawn Submission_

### Official Review · Reviewer_pcfq · 2024-10-31

**Soundness:** 3
**Presentation:** 3
**Contribution:** 3
**Rating:** 6
**Confidence:** 4

**Summary:**

This paper focuses on online dynamic rendering under a multi-view camera setting. The proposed method, LayeredGS, could achieve high-quality rendering and fast convergence. The key to achieving this is a multi-level, coarse-to-fine training strategy based on K-Means clustering. Through several experimental evaluations, this method has demonstrated to be efficient.

**Strengths:**

● The proposed multi-level clustering-based learning strategy is both reasonable and efficient, demonstrating that the incorporation of structural information can greatly aid motion learning.

● The article provides sufficient ablation experiments for the proposed modules, which help readers understand in detail the impact of each specific strategy on optimization.

● The article is clearly written and easy to read.

**Weaknesses:**

● This paper primarily compares only three papers—Dynamic3DGS, SC-GS, and RealTime4DGS—while there are other online dynamic scene reconstruction methods that should be considered, such as 3DGSstream, Neural Residual Radiance Fields for Streamably Free-Viewpoint Videos, and Streaming Radiance Fields for 3D Video Synthesis, etc..

● The dataset used for validation in the paper features relatively simple scenes and clear motion. The paper could consider conducting further validation experiments on the Nvidia Dynamic Scene Dataset, N3DV, and UCSD datasets.

● In the contribution summary, the paper mentioned “We demonstrate one novel application, Self-supervised Articulated Object Segmentation, showcasing the utility of deformation information for the first time.”. I don’t think this is the first time the deformation utility demonstrated, there are lots of works doing scene decoupling based on learned motion, like Dˆ2neRF, and NeuralDiff. Also, there are some works that focus on the “self-supervised articulated object segmentation” mentioned in the paper, like Watch-It-Move and MovingParts, but these works are not discussed in the paper.

**Questions:**

● The multi-layer clustering structure in the paper relies on Gaussian center positions, which tends to result in relatively simple shapes for each cluster. When the number of clusters is limited and the scene is complex, could this cause the method to fail?

● From Equation 9, it can be seen that the deformation of each point is obtained by stacking multiple layers of deformation functions. How can we demonstrate that each layer serves its intended role, such as lower-level functions modeling coarse-grained motion and higher-level functions capturing local deformations, rather than having one layer dominate the overall effect?

● How many viewpoints are used for training each frame in the paper? Would reducing the number of camera viewpoints lead to failure?

---

### Official Review · Reviewer_3kKN · 2024-11-02

**Soundness:** 2
**Presentation:** 2
**Contribution:** 2
**Rating:** 3
**Confidence:** 3

**Summary:**

The authors present LayeredGS, a novel approach for simultaneous novel view synthesis and point tracking in dynamic scenes. Specifically, LayeredGS can reconstruct the 3D scene in an "online" manner, i.e., in a frame-by-frame fashion.

LayeredGS first clusters Gaussians in the scene into several clusters. This clustering is performed in a hierarchical style with three layers in total. Then, the deformation of each cluster between frames is modeled as a parameterized affine transformation with tanh-activated scaling coefficients. The final deformation of a Gaussian is generated by traversing and nesting the deformations of all its parent clusters. Extra residual parameters are also added to fine-tune the final deformed Gaussians.

The authors evaluate LayeredGS on the Panoptic dataset and an accelerated Particle-NeRF dataset, with an extra constraint of 100 iterations per frame training. Results show that the proposed method surpasses previous methods in terms of rendering quality and 2D tracking accuracy. Furthermore, the training time per frame is less than 3 seconds.

**Strengths:**

- The hierarchical clustering architecture effectively reduces the optimization dimensionality by organizing numerous Gaussians into manageable sets.

- The method demonstrates rapid convergence for dynamic scenes with substantial motion, outperforming existing approaches that typically require longer convergence times.

- The achieved processing time (1-3 seconds per frame) shows promising potential for near-real-time dynamic scene reconstruction applications.

**Weaknesses:**

- The contribution of the paper can be summarized in one sentence: using a fixed yet inflexible method to enforce the as-rigid-as-possible regularization, which has already been widely used in other dynamic 3DGS projects including SCGS and HiFi4G. Previous methods have abundant mechanisms to grow, prune, and adjust anchor points, while the proposed method does not present such flexibility for Gaussian clustering. This dramatically undermines the generalizability of the method.

- The clustering criterion is simply the Gaussians' location, which in most cases could not provide enough guarantee of similar deformation behavior, let alone the same affine deformation behavior. This is also why previous works adopt rigidity as a regularization term instead of a hard constraint.

- The multilayer design seems unnecessary. Equation (8) itself is simply an affine transformation (assuming there is no scaling in the scene, which widely holds in your evaluated dataset). And the nested multi-layer affine is still an affine. Since the benefits of reducing the optimization space have already been achieved by the finest layer of clustering, the motivation for multiple layer structure is questionable.

- The evaluation methodology raises several concerns:
  - Throughout the comparison, the authors claimed that "for fairness" they allow other methods to train for merely 100 iterations per new frame. Obviously, most other methods do not achieve model convergence, and the comparison in rendering quality in this case is meaningless. A better comparison setup would be the average iterations it takes for those models to converge as well as the final metrics after convergence. Time to converge could also be a very informative metric as long as you are running the experiment under the same hardware. All these projects are open-sourced, so you are not expected to worry about implementation variance.
  - The customized "accelerated dataset" is confusing, especially why reducing the number of frames in Particle-NeRF is necessary in the evaluation setup. The vanilla Dynamic3DGS could achieve a 39 PSNR on the Particle-NeRF dataset; however, in your setup, it only achieves less than 30 PSNR. While I understand your intention to magnify the advantages of your method, at least show the results of the original dataset so that people can judge your work fairly.
  - The exclusion of DNeRF and HyperNeRF datasets citing "online case" limitations appears selective, particularly given the existence of other online monocular reconstruction methods like DynOMo(https://arxiv.org/pdf/2409.02104).
  - The omission of N3D and Immersive datasets due to "small movement scale" seems unjustified. If these datasets are indeed simpler cases, demonstrating superior performance would strengthen the method's validation.

**Questions:**

- Regarding 2D tracking, why utilize manual keypoint annotations when Dynamic3DGS employs automated facial and hand keypoint detection methods?
- Please clarify the implementation details for K=1 in Table 4, specifically regarding cluster count.
- In Equation (8), what justifies using cluster centroids as rotation/scaling centers, given that optimal transformation centers may lie elsewhere?
- The clustering arrow direction in Figure 2 appears reversed relative to Equation 9's formulation. It should be from finer cluster to coarse cluster.
- The pendulum trajectories in Figure 4 exhibit unexpected curvature at their termini, deviating from expected arc-like paths. Could you explain this behavior?
- A comparative analysis of model sizes would be valuable, particularly given the additional parameters introduced by the residual block for each Gaussian.
- Please explain the rationale behind the different camera configurations used for ParticleNeRF (36+4) versus Dynamic3DGS (20 train, 10 test).

---

### Official Review · Reviewer_6xTp · 2024-11-03

**Soundness:** 2
**Presentation:** 2
**Contribution:** 2
**Rating:** 5
**Confidence:** 4

**Summary:**

This paper proposed an efficient online 4D Gaussian splatting method for multi-view deformable scenes. LayeredGS optimized the efficiency for 4DGS by grouping Gaussian with similar deformation fields. They also proposed a multi-layer cluster strategy for a more efficient training procedure.

**Strengths:**

The introduction of motivation is clear and easy to understand.

This paper provided sufficient experimental results on two mult-view dynamic scene datasets.

**Weaknesses:**

1. The introduction of Dynamic Novel-View Synthesis in the related works section is redundant. To be specific, it's better to focus on the difference between your method with other concurrent methods. Static novel view synthesis is not necessary since your paper is focus on dynamic scene reconstruction. Though this paper highlighted the difference between their method and SCGS in lines 153 to 158, the high-level idea of those two methods is the same, which are k-means cluster. The main difference is this paper learns clusters for each cluster, which is called multi-layer clusters in the paper.

2. It is not clear what exactly the loss function LayeredGS applied. Moreover, it is not clear how to formulate each Gaussian or cluster across time. As introduced in line 277, LayeredGS optimized $R_j$, $t_j$, $c_j$, and $s_j$. Do you need to optimize $R_j$, $t_j$, $c_j$, and $s_j$ for each frame? Besides, since the movement of each Gaussian is with respect to its corresponding cluster, how do you formulate the deformation of each cluster exactly? Generally speaking, this part is so confusing for me. Could you pls further provide:

- A clear formulation of the loss function used in LayeredGS.
- A detailed explanation of how Gaussians and clusters are formulated across time.
- Clarification on whether $R_j$, $t_j$, $c_j$, and $s_j$ are optimized for each frame.
- A precise formulation of how the deformation of each cluster is calculated.
- The detailed loss function and the optimization procedure.

3. Since 3DGStream and SpacetimeGS [1] are developed for multi-view 4DGS rendering, it is more convincing to compare LayeredGS with those two methods on the Neural 3D Video dataset and meet-room dataset, even though SpacetimeGS is an offline method. You may consider including comparisons with 3DGStream and SpacetimeGS on the suggested datasets, or explain why such comparisons were not included with convincing reasons.

4. The comparisons with other methods are not fair. First, 4DGS and SCGS are mainly designed for monocular dynamic scene rendering. Though 4DGS showed the performance on the Neural 3D Video dataset, the performance is still lower than 3DGStream or SpacetimeGS since 3DGStream and SpacetimeGS are designed for multi-view deformable scene reconstruction. Second, as mentioned in lines 407 to 417, the total iteration is set to 100 multiplied by the total number of frames for other compared methods. I'd say the total iteration is insufficient for those compared methods to reach their optimal performance. You can get better performance for compared methods by increasing the training period. Could you further:

- Provide a clearer justification for including monocular methods in their comparison, or focus more on multi-view online or offline methods.
- Run additional experiments with increased training iterations for the compared methods to ensure they reach optimal performance.
- Include a discussion of the limitations of their current comparison approach and how it might affect the interpretation of the results.

[1] Li, Zhan, et al. "Spacetime gaussian feature splatting for real-time dynamic view synthesis." Proceedings of the IEEE/CVF Conference on Computer Vision and Pattern Recognition. 2024.

**Questions:**

1. Fig 8 shows the comparison of the convergence speed for Dynamic3DGS and LayeredGS. How's the convergence speed and training time for 3DGStream?

2. Fig V in the supp shows the cluster for each layer. Could you show more cluster structures on other real-world datasets? For instance, Neural 3D Video dataset and Panoptic dataset.

3. How's the comparison with other multi-view 4DGS methods, for instance, 3DGStream and SpacetimeGS?

4. How's the performance for 4DGS, Deformable3DGS, and SC-GS with sufficient training iteration?

5. This paper claims in line 150 that a multi-level structure for 3DGS makes online training more efficient. Why? Is there any experimental result to prove this? SCGS also learns clusters for Gaussians, but the training speed is not faster than 4DGS or other methods. Could you provide a more detailed comparison of your multi-layer clustering approach to SCGS's single-layer clustering, highlighting the key advantages of your proposed method?

---

### Official Review · Reviewer_kNjJ · 2024-11-04

**Soundness:** 2
**Presentation:** 2
**Contribution:** 2
**Rating:** 5
**Confidence:** 5

**Summary:**

LayeredGS introduces a novel method for dynamic scene rendering and point tracking using multi-layer deformable Gaussian splatting. By grouping Gaussians with similar deformations into hierarchical layers and employing a trainable deformation function for clusters, LayeredGS achieves significant optimization efficiency. This approach requires only 100 iterations per frame—substantially fewer than previous methods—while maintaining high rendering quality and accurate 3D tracking of scene elements. Additionally, LayeredGS facilitates self-supervised articulated object segmentation, demonstrating its utility in capturing complex motions. Experimental results show that it achieves performance comparable to state-of-the-art methods with a fraction of the training time.

**Strengths:**

The strengths of this paper lays in the following several aspects.

1. The paper is well-organized and clearly presents the motivation, methodology, and results.

2. The proposed method demonstrates improvements in training efficiency, achieving rapid convergence within 100 iterations per frame on fast-moving dynamic datasets. This is a substantial enhancement compared to previous methods requiring up to 2000 iterations per frame.

3. The paper introduces LayeredGS, a novel method for dynamic scene rendering and point tracking that utilizes multi-layer deformable Gaussian splatting. This approach extends the capabilities of 3D Gaussian Splatting (3DGS) to dynamic scenes, which is an innovative contribution to the field.

4. By leveraging a multi-layer coarse-to-fine clustering structure, the method effectively captures deformations in dynamic scenes, which is a creative combination of Gaussian splatting with hierarchical clustering techniques.

**Weaknesses:**

The weakness of this paper lays in the following aspects. My final decision will consider the feedback of the authors during the discussion section and I would like to raise the score if some of the major concerns are solved.

1. Compared with the data-sets adopted in the DynamicGS, the comparative experiments are small and limited. The evaluation is primarily conducted on the Panoptic and FastParticle datasets. While these datasets are challenging, the paper would benefit from testing on a broader range of datasets, including more real-world scenarios, to demonstrate the generalizability of the method.

2. The paper lacks comprehensive ablation studies to quantify the contributions of different components of the proposed method. For instance, evaluating the impact of the number of layers in the clustering structure or the choice of clustering algorithms would provide deeper insights.

3. Some aspects of the multi-layer clustering and deformation optimization are not fully detailed. Providing pseudo-code or algorithmic summaries in the main text could enhance reproducibility.
More explanations on how clusters are merged across layers and how this affects the deformation modeling would improve clarity.

**Questions:**

See the weakness section.

---

### Note · Authors · 2024-11-14

I have read and agree with the venue's withdrawal policy on behalf of myself and my co-authors.